# Emerging Role of Targeted Therapy in Metastatic Pancreatic Adenocarcinoma

**DOI:** 10.3390/cancers14246223

**Published:** 2022-12-16

**Authors:** Brandon M. Huffman, Haley Ellis, Alexander C. Jordan, William A. Freed-Pastor, Kimberly Perez, Douglas A. Rubinson, Nilay Sethi, Harshabad Singh, Rishi Surana, Brian M. Wolpin, Andrew J. Aguirre, James M. Cleary

**Affiliations:** 1Division of Gastrointestinal Oncology, Department of Medical Oncology, Dana-Farber Cancer Institute, Harvard Medical School, Boston, MA 02215, USA; 2Division of Molecular and Cellular Oncology, Department of Medical Oncology, Dana-Farber Cancer Institute, Harvard Medical School, Boston, MA 02215, USA

**Keywords:** pancreatic adenocarcinoma, precision medicine, targeted therapy, DNA repair inhibitors

## Abstract

**Simple Summary:**

Metastatic pancreatic ductal adenocarcinoma (PDAC) is an aggressive malignancy that has limited treatment options. Standard of care treatment involves systemic chemotherapy, although all tumors invariably develop resistance to these cytotoxic therapies. With the advent of genomic sequencing and identification of therapeutically actionable alterations, there are subsets of patients with PDAC who may benefit from targeted therapies matched to their molecular profile. As more molecularly matched therapies are developed, precision medicine has great potential in patients with PDAC.

**Abstract:**

The aggressive biology of pancreatic ductal adenocarcinoma (PDAC), along with its limited sensitivity to many systemic therapies, presents a major challenge in the management of patients with metastatic PDAC. Over the past decade, the incorporation of combinatorial cytotoxic chemotherapy regimens has improved patient outcomes. Despite these advances, resistance to cytotoxic chemotherapy inevitably occurs, and there is a great need for effective therapies. A major focus of research has been to identify molecularly defined subpopulations of patients with PDAC who may benefit from targeted therapies that are matched to their molecular profile. Recent successes include the demonstration of the efficacy of maintenance PARP inhibition in PDAC tumors harboring deleterious *BRCA1*, *BRCA2*, and *PALB2* alterations. In addition, while therapeutic targeting of *KRAS* was long thought to be infeasible, emerging data on the efficacy of *KRAS* G12C inhibitors have increased optimism about next-generation *KRAS*-directed therapies in PDAC. Meanwhile, *KRAS* wild-type PDAC encompasses a unique molecular subpopulation of PDAC that is enriched for targetable genetic alterations, such as oncogenic *BRAF* alterations, mismatch repair deficiency, and *FGFR2*, *ALK*, *NTRK*, *ROS1*, *NRG1*, and *RET* rearrangements. As more molecularly targeted therapies are developed, precision medicine has the potential to revolutionize the treatment of patients with metastatic PDAC.

## 1. Introduction

Pancreatic ductal adenocarcinoma (PDAC) is the third leading cause of cancer-related death in the United States, and is projected to become the second leading cause by 2030 [1]. The 5-year survival rate of PDAC patients is only 11%, and the poor outcomes are mainly due to its aggressive biology and advanced stage of presentation at initial diagnosis [2]. The cornerstone of treatment for this lethal disease is cytotoxic chemotherapy, and the current standard of care is largely based on clinical trials of first-line therapy initially reported in 2011 and 2013 [3,4,5,6,7,8]. In this review, we describe the current treatment paradigms for metastatic PDAC and how precision medicine is beginning to improve patient outcomes by leveraging the increased identification of molecularly-defined populations of PDAC.

## 2. First-Line Chemotherapy

Outcomes for patients with PDAC have improved over the last three decades with the development of effective cytotoxic chemotherapy regimens [6,7,9]. Initially, the median overall survival (OS) of patients with advanced PDAC treated with gemcitabine alone was 5.7 months, setting an ominous benchmark for future generations to improve upon [10]. Compared to gemcitabine monotherapy, multi-drug cytotoxic chemotherapy regimens have significantly improved outcomes by prolonging the OS of patients with metastatic PDAC to about 1 year. In the landmark PRODIGE trial, patients with metastatic PDAC treated with first-line FOLFIRINOX (folinic acid, 5-fluorouracil [5-FU], irinotecan, and oxaliplatin) had a median OS of 11.1 months, compared to 6.8 months for gemcitabine alone [3]. In the MPACT trial, gemcitabine/nab-paclitaxel prolonged survival to 8.5 months, compared to 6.7 months with gemcitabine alone, in previously untreated patients with advanced PDAC [4].

While the results of head-to-head investigation of FOLFIRINOX and gemcitabine/nab-paclitaxel have not been reported, many believe that FOLFIRINOX may be more potent, as cross-trial comparisons suggest it may have a higher objective response rate (31.6% versus 23%). To date, the highest reported median OS for an unselected advanced PDAC population was in the control arm of SWOG 1313, a randomized phase II trial assessing FOLFIRINOX with and without pegylated recombinant human hyaluronidase (PEGPH20). The median OS was 14.4 months in the FOLFIRINOX arm, while it was 7.7 months in the arm combining FOLFIRINOX and PEGPH20 arm [11].

The American Society of Clinical Oncology (ASCO) Clinical Practice Guidelines have endorsed the use of either FOLFIRINOX or gemcitabine/nab-paclitaxel as first-line therapy in patients with a favorable comorbidity profile and ECOG performance status of 0 or 1 [12,13,14]. The toxicity profiles of these two regimens differ significantly; therefore, patient-centered decision making is necessary [15]. While it may be more potent, FOLFIRINOX is a more challenging regimen, with a higher rate of grade 3–4 toxicities and hospitalizations. The more favorable toxicity profile of gemcitabine/nab-paclitaxel leads some patients with substantial comorbidities or desire to prioritize quality of life to elect to be treated with gemcitabine/nab-paclitaxel [16].

In an effort to maximize the tolerability and efficacy of first-line therapy, the recently reported SEQUENCE trial used a novel strategy of alternating gemcitabine/nab-paclitaxel with 5-FU, folinic acid and oxaliplatin (FOLFOX) [17]. In this randomized phase III study, alternating gemcitabine/nab-paclitaxel and FOLFOX was compared to standard gemcitabine/nab-paclitaxel in patients with previously untreated metastatic PDAC. The trial met its primary endpoint of improving OS. Patients receiving alternating gemcitabine/nab-paclitaxel and FOLFOX had a median OS of 13.2 months versus 9.7 months in patients receiving standard gemcitabine/nab-paclitaxel. The ORR for the alternating combination was 39.7% versus 20.3% for gemcitabine/nab-paclitaxel. Notably, there were significantly more hematologic toxicities in patients receiving alternating gemcitabine/nab-paclitaxel and FOLFOX, along with a higher incidence of treatment-emergent neuropathy. The results from this trial suggest that alternating regimens may slow the development of acquired chemotherapy resistance; however, this approach needs further study before it can be adopted into the current treatment paradigm for metastatic PDAC.

## 3. Second-Line Chemotherapy

Several randomized phase III trials have investigated second-line cytotoxic chemotherapy regimens after progression on first-line gemcitabine-based chemotherapy. The use of 5-FU and oxaliplatin has shown mixed results. The CONKO-003 evaluated second-line oxaliplatin, 5-FU, and folinic acid (OFF regimen) versus 5-FU/Leucovorin in advanced PDAC [18]. Compared to patients receiving second-line 5-FU/Leucovorin, patients receiving OFF experienced improved OS (median OS 5.9 months versus 3.3 months) and PFS (median PFS 2.9 months versus 2.0 months). However, in the PANCREOX trial investigating a different dosing regimen of second-line folinic acid, 5-FU, and oxaliplatin (mFOLFOX6) versus 5-FU/Leucovorin, the primary endpoint of PFS was not met (median PFS 3.1 months versus 2.9 months, respectively), and median OS was actually reduced in patients treated with mFOLFOX6 compared to patients treated with 5-FU/Leucovorin (6.1 months versus 9.9 months) [19]. Recent clinical trials using second-line mFOLFOX6 as a comparator arm have demonstrated consistent outcomes, with median PFS of 2–3 months and median OS of 6–7 months [20,21].

Trials evaluating second-line irinotecan-containing regimens have demonstrated similar levels of modest efficacy. The NAPOLI-1 trial evaluated nanoliposomal irinotecan (Nal-IRI) combined with 5-FU/Leucovorin, compared to 5-FU/Leucovorin alone. Nal-IRI/5-FU was superior to 5-FU/Leucovorin, improving median OS (6.1 months versus 4.2 months), median PFS (3.1 months versus 1.5 months), and the ORR (16% versus 1%) [22]. A randomized phase II trial, SWOG S1513, assessed whether the addition of the poly (adenosine diphosphate-ribose) polymerase (PARP) inhibitor veliparib would enhance the effectiveness of second-line FOLFIRI in metastatic PDAC. While SWOG S1513 was stopped early for futility, because veliparib did not increase the effectiveness of FOLFIRI, results from this trial suggest that FOLFIRI has similar activity to Nal-IRI/5-FU. Patients treated on the FOLFIRI arm of SWOG S1513 had a median PFS of 2.9 months and a median OS of 6.5 months [23].

Contrary to the large number of published phase III trials assessing therapies after progression while on gemcitabine-based therapy, there are relatively few studies on the use of therapies after progression on 5-FU based therapies [15,24,25]. However, emerging data suggest that the efficacy of second-line gemcitabine/nab-paclitaxel is very similar to the efficacy of second-line 5-FU based regimens. Although it was a negative trial, a randomized phase II trial investigating the use of second-line gemcitabine/nab-paclitaxel with and without a MUC5AC antibody provided efficacy benchmarks of second-line gemcitabine/nab-paclitaxel [25]. In this trial, the median OS for patients receiving second-line gemcitabine/nab-paclitaxel alone was 6.6 months; the median PFS was 2.7 months; and the ORR was 3% [25]. The phase III Trybeca-1 trial, which evaluated second-line chemotherapy (investigator choice) with or without eryaspase, had similar efficacy results. In the Trybeca-1 trial, patients who received second-line gemcitabine/nab-paclitaxel had a median OS of 6.9 months and median PFS of 3.5 months [24].

## 4. Biomarker-Driven Therapy

Given the impressive results of precision medicine using molecularly targeted therapy in other malignancies, there have been significant efforts to target oncogenic alterations in PDAC [26]. However, a major obstacle has been that the most frequently observed molecular alterations in PDAC have historically been undruggable. More than 90% of PDACs harbor an oncogenic Kirsten rat sarcoma virus (*KRAS*) mutation [27]. Other common oncogenic alterations in PDAC include tumor suppressor genes, such as *CDKN2A*, *TP53*, and *SMAD4* [27,28]. While this mutational profile presents a clinical challenge, optimism for targeted therapy in PDAC is rising with the identification of molecularly defined subpopulations of PDAC that appear to be sensitive to targeted therapies. For instance, while *KRAS* is notoriously difficult to target, PDACs lacking *KRAS* mutations (*KRAS* wild type [WT] PDAC) frequently harbor targetable alterations, such as *NTRK*, *FGFR2*, *ALK*, *RET*, *NRG1*, and *ROS1* fusions, as well as oncogenic *BRAF* alterations (Figure 1).

In addition, there is significant enrichment of mismatch repair deficiency in KRAS WT PDAC [29]. Beyond *KRAS* WT PDAC, approximately 10% of PDACs have a deficiency in the homologous recombination (HR) DNA repair pathway, with inactivating germline or somatic mutations in *BRCA1*, *BRCA2*, and *PALB2* [27]. The “Know Your Tumor” platform identified targetable alterations in 26% of PDAC tumors [31]. In a retrospective analysis, those who received molecularly matched targeted therapy had improved OS (median OS 2.6 years versus 1.5 years) compared to those who received unmatched therapies. Many of these targeted therapies hold FDA approval for tumors other than PDAC (Table 1). These data support the hypothesis that precision medicine has a role in the treatment of PDAC and should continue to be validated prospectively.

### 4.1. Homologous Recombination Deficiency

Double-strand DNA breaks are repaired by the homologous recombination (HR) repair pathway, and deficiency of this DNA repair pathway confers sensitivity to platinum chemotherapy and PARP inhibitors in multiple cancers [32,33]. Since approximately 17 genes are involved in the HR pathway, it is helpful to distinguish between core HR genes (*BRCA1*, *BRCA2*, *PALB2*) and the other non-core HR genes. Deleterious alterations in core HR genes are well-established markers of sensitivity to PARP inhibitors and platinum chemotherapy. Alternatively, while deleterious alterations in some non-core HR genes, like RAD51C, have been reported to impart PARP inhibitor sensitivity in ovarian cancer, the impact of other non-core HR genes is less clear [34]. Deleterious alterations in core HR genes occur in 10–12% of PDAC patients, while deleterious alterations in non-core HR genes are found in approximately 7% of PDACs [27,35]. Multiple studies have demonstrated that PDAC patients with mutations in a core HR gene have enhanced sensitivity to platinum chemotherapy, leading to improved PFS, compared to HR-proficient PDACs, when they are treated with first-line platinum-based chemotherapy [31,36,37,38].

A phase II trial evaluating gemcitabine/cisplatin with or without the PARP inhibitor veliparib in PDAC patients with germline *BRCA1*, *BRCA2*, or *PALB2* mutations highlights the enhanced platinum-sensitivity in HR-deficient PDAC [39]. While veliparib did not improve the effectiveness of first-line gemcitabine/cisplatin, the trial clearly demonstrated the impressive efficacy of gemcitabine/cisplatin in HR-deficient PDAC. Patients treated in this trial with gemcitabine/cisplatin achieved a median OS of 16.4 months, median PFS of 9.7 months, and an ORR of 65.2%. Interestingly, in a separate trial studying an unselected population of patients with PDAC, gemcitabine/cisplatin did not statistically improve OS or ORR compared to gemcitabine alone [40]. There are no trials comparing gemcitabine/cisplatin to FOLFIRINOX in HR-deficient PDAC, although either is reasonable in PDAC tumors with deleterious mutations in *BRCA1*, *BRCA2*, or *PALB2*.

PARP inhibitors have shown significant efficacy in multiple HR-deficient tumors, with a toxicity profile more favorable than cytotoxic chemotherapy [41,42]. An important lesson from other malignancies, which has also proven true in PDAC, is that PARP inhibitors are only efficacious in platinum-sensitive tumors. This insight led to the design of the POLO trial, where maintenance olaparib versus placebo was studied in patients with PDAC with deleterious germline *BRCA1* or *BRCA2* mutations who had no evidence of disease progression following at least 16 weeks of platinum-based chemotherapy. The trial met its primary endpoint of improving PFS (7.4 months versus 3.8 months, olaparib versus placebo), but did not demonstrate a statistically significant improvement in OS [43,44]. On the basis of the POLO trial, the FDA approved maintenance olaparib in PDAC patients with germline BRCA mutations. However, a major question in the field is whether other PDAC patients can benefit from maintenance PARP inhibition. A phase II trial evaluating maintenance rucaparib in PDAC with deleterious germline or somatic *BRCA1*, *BRCA2*, and *PALB2* mutations who did not progress on first-line platinum-based chemotherapy also demonstrated encouraging results, with a median PFS of 13.1 months [45]. These results suggest that maintenance PARP inhibition is a reasonable therapeutic strategy in this patient population [45,46]. Interestingly, a recent study demonstrated impressive efficacy of a maintenance strategy using the CTLA4 inhibitor ipilimumab combined with the PARP inhibitor niraparib in patients who had not progressed on platinum-based chemotherapy and tumors without an underlying *BRCA1/2* or *PALB2* mutation [47]. This may prove to be an effective maintenance strategy, but further investigation is needed.

### 4.2. KRAS Mutated PDAC

While the *KRAS* gene is mutated in approximately 14% of all human cancers, PDAC is the prototypical *KRAS* mutated malignancy, as over 90% of PDACs harbor a *KRAS* mutation [48]. *KRAS* is a GTPase that activates downstream effector proteins of the mitogen-activated protein kinase (MAPK) pathway [49]. Genomic studies investigating the frequency of different *KRAS* mutated alleles in PDAC have demonstrated the following distribution of *KRAS* mutations: G12D (41%), G12V (31%), G12R (14%), Q61H (4%), and G12C (<1%) [27,30]. Translational studies exploring patient outcomes and co-mutational patterns have begun to reveal differences amongst tumors harboring different *KRAS* mutation alleles (Figure 1) [50,51]. Patients with PDAC harboring a *KRAS* G12D mutation have decreased OS compared to PDAC patients with mutations in other *KRAS* alleles. Interestingly, PDAC with *KRAS* G12R mutations are more likely to harbor deleterious HR gene alterations and PI3-kinase pathway mutations [52,53].

In an encouraging step forward, *KRAS* G12C has been successfully targeted in PDAC patients with the direct irreversible *KRAS* G12C inhibitors sotorasib and adagrasib [54,55]. The CodeBreaK100 study was an open-label phase I/II clinical trial investigating the role of sotorasib in previously treated solid tumors harboring *KRAS* G12C mutations, including 38 patients with metastatic PDAC [54]. In the PDAC cohort, the ORR was 21%; the disease control rate (DCR) was 84%; the median PFS was 4.0 months; and the median OS was 6.9 months [54]. The drug was well-tolerated, with only 5.3% of patients experiencing grade 3 diarrhea or fatigue. The KRYSTAL-1 study was a multicohort phase I/II clinical trial investigating the role of adagrasib, another covalent irreversible *KRAS* G12C inhibitor, in advanced gastrointestinal cancers [55]. In a preliminary analysis of 10 evaluable PDAC patients enrolled on KRYSTAL-1, remarkably, 5 PDAC patients achieved a partial response, with a median duration of response of 7 months, and the other five patients had stable disease, yielding a median PFS of 6.6 months [55]. Overall, adagrasib was well to-erated. Gastrointestinal toxicity was the most common adverse event, while fatigue and QTc prolongation were the most common grade 3–4 toxicities.

There are currently no FDA-approved *KRAS* inhibitors available for patients with PDAC outside a clinical trial. However, the success of *KRAS* G12C inhibitors has inspired optimism that KRAS-directed therapy in PDAC is feasible, and investigational agents, such as KRAS G12D inhibitors, SOS inhibitors, and pan-RAS inhibitors, are being actively developed. Encouraging preclinical data have been presented on the KRAS G12D inhibitor MRTX1133 and the pan-RAS inhibitor RMC-6236 [56,57]. MRTX1133 is a non-covalent small molecule inhibitor that binds to GDP-loaded *KRAS* G12D and leads to impressive anti-tumor activity in *KRAS* G12D-mutated murine models [57]. RMC-6236 inhibits all RAS isoforms (i.e., *K*-, *H*-, and *NRAS*) by acting as a molecular glue to promote the binding RAS proteins to the cyclophilin A chaperone protein. Stable formation of the RAS/cyclophilin A complex inhibits the ability of RAS to activate downstream MAPK effectors. The RMC-6236 pan-RAS inhibitor demonstrated striking anti-tumor activity, with several complete responses, in murine *KRAS*-mutated xenografts [56].

In addition to targeted therapy approaches, immunotherapeutic strategies utilizing *KRAS*-directed vaccines and T-cell directed therapies are also being developed [28,58]. The potential of this approach was highlighted in a case report of a patient with *KRAS* G12D-mutated PDAC, who had a lung-only metastasis and experienced a partial response (72% tumor regression) following infusion of autologous T cells that were engineered to express HLA-C*08:02–restricted T-cell receptors (TCRs) targeting oncogenic *KRAS* G12D [59]. This demonstration of TCR-directed therapy in PDAC highlights the potential for broader application of immunotherapeutic approaches in this cancer, which has historically been resistant to immunotherapy [60].

### 4.3. KRAS Wild-Type (WT) PDAC

Given the historic difficulties in targeting *KRAS*, increasing attention has been given to *KRAS* WT PDAC [29,61]. In addition to being enriched for mismatch repair-deficient tumors, multiple reports have demonstrated that *KRAS* WT PDAC can harbor oncogenic *BRAF* alterations and gene fusions of *ALK*, *RET*, *ROS1*, *NRG1*, *FGFR2*, and *NTRK* [29,61,62,63]. *KRAS* WT PDAC is more common in younger patients and appears to have longer OS compared to patients with *KRAS*-mutated PDAC [29,61]. Interestingly, genomic studies have also observed that *KRAS* WT PDAC has a higher frequency of cancers that are *ATM*-mutated and *TP53* WT [61,64,65]. However, given the rare incidence of *KRAS* WT PDAC, data regarding therapeutic targeting in patients is sparse.

The NOTABLE study provided proof-of-concept that *KRAS* WT PDAC is susceptible to molecularly targeted therapy [66]. The NOTABLE study was a randomized, double-blinded, phase III clinical trial conducted in China that investigated the role of gemcitabine combined with nimotuzumab, a humanized anti-epidermal growth factor receptor (EGFR) antibody, compared to gemcitabine monotherapy as first-line treatment for KRAS WT advanced PDAC [66]. Median OS was significantly improved at 10.9 months with gemcitabine/nimotuzumab compared to 8.5 months with gemcitabine alone. Median PFS was 4.2 months with gemcitabine/nimotuzumab compared to 3.6 months with gemcitabine alone. Clinically, although there was a slight improvement in OS in the NOTABLE study, it is difficult to put the results of this trial into context. since it did not include a modern chemotherapy regimen, such as FOLFIRINOX or gemcitabine/nab-paclitaxel.

### 4.4. BRAF Alterations

Activating mutations in *BRAF* activate downstream signaling in the MAPK pathway and targeting *BRAF* V600E class 1 mutations with *BRAF*-targeted therapy has proven effective in multiple malignancies [27,67,68]. In a report analyzing PDAC and pancreatic acinar cell carcinomas, oncogenic *BRAF* alterations accounted for 13% of the *KRAS* WT tumors [29]. Differing from the experience in melanoma, *BRAF* V600E mutations only accounted for 20–30% of the *BRAF* alterations seen in PDAC and pancreatic acinar cell carcinomas [27,29,65]. Other *BRAF* alterations included *BRAF* exon 11 insertion-deletion mutations (INDELs) and *BRAF* fusions. Clinical experience targeting *BRAF* alterations in pancreatic cancer is limited [27]. In a retrospective case series of PDAC and acinar cell carcinoma, two patients with *BRAF* V600E mutations achieved a partial response on combined *BRAF*/*MEK* inhibition [69]. In the same cohort of patients, there were responses to *MEK* inhibitor monotherapy in the *BRAF* fusion and *BRAF* INDEL subgroups, but survival outcomes were not reported due to the uncontrolled nature of the case series [27].

### 4.5. FGFR2 Fusions

Fibroblast growth factor receptor-2 (*FGFR2*) fusions account for 3.5% of the genomic alterations *KRAS* WT PDAC and pancreatic acinar cell carcinoma [29,70]. *FGFR2* receptors typically become activated when its ligand, fibroblast growth factor, promotes *FGFR2* dimerization. Fusion partners in *FGFR2* translocations have intracellular dimerization domains that promote constitutive *FGFR2* activation, resulting in downstream activation of the MAPK pathway [71,72]. While therapeutic targeting of *FGFR2* translocations has been efficacious in intrahepatic cholangiocarcinoma, clinical experience in targeting of these alterations in PDAC is very limited. In the FIGHT-101 trial, a patient with *KRAS* WT PDAC harboring an *FGFR2-USP33* fusion achieved a partial response to pemigatinib that was maintained for 10.7 months before disease progression [73]. A recent case series described four *KRAS* WT PDAC patients with *FGFR2* fusions who benefited from an undisclosed *FGFR2* inhibitor on an unpublished clinical trial [74]. Three of the patients with *KRAS* WT PDAC benefited from the use of an FGFR inhibitor for at least 6 months [74]. The most common side effects with FGFR inhibitors are hyperphosphatemia, cutaneous toxicities, and ocular toxicities, although these toxicities are manageable with dose and schedule modifications.

### 4.6. ALK Fusions

Anaplastic lymphoma kinase fusion (*ALK*) gene-rearrangement proteins promote tumor growth and cell survival by activating intracellular signaling cascades, including the MAPK pathway [75,76]. Therapeutic targeting of *ALK* fusions with small molecule inhibitors has been successful in other malignancies, such as lung cancer and anaplastic large cell lymphoma [77,78,79,80]. *ALK* fusions are seen in approximately 2.5% of *KRAS* WT PDAC, and recent reports have shown promising efficacy of *ALK* inhibitors in *ALK*-rearranged PDAC [77,81,82]. A case series of *KRAS* WT metastatic PDAC harboring *ALK* rearrangements described four patients who benefitted from *ALK*-directed therapy, with two of these patients being on therapy for over 1 year [83]. In a separate case series, five patients with *KRAS* WT PDAC harboring *ALK* rearrangements received *ALK*-directed therapy [84]. In this case series, a response to therapy was seen in one PDAC patient treated with crizotinib and one PDAC patient treated with alectinib [84]. Treatment with *ALK* inhibitors is generally well-tolerated, although gastrointestinal and pulmonary toxicities have been reported [78,82,85].

### 4.7. NTRK Fusions

The tropomysin receptor kinase (TRK) family of receptors are stimulated by binding to their ligand, neurotrophins, causing receptor dimerization and activation of downstream signaling pathways [86]. *NTRK* fusions of the genes that code for TRK receptors (*NTRK1-3*) encode proteins that lead to dimerization and constitutive receptor activity [86]. *NTRK* fusions occur in 1–3% of *KRAS* WT PDAC. In a pooled analysis of three tumor agnostic clinical trials for patients with *NTRK* fusions, there were two patients with *NTRK*-rearranged PDAC, and one of these patients had a partial response to the *NTRK* inhibitor larotrectinib [87]. In another case report, a *KRAS* WT PDAC patient with a *CTRC-NTRK1* fusion responded to larotrectinib for 6 months prior to developing acquired resistance [88]. In a separate meta-analysis of three clinical trials investigating the *NTRK* inhibitor entrectinib, three patients with PDAC harboring *NTRK* rearrangements were treated; two of these achieved a partial response [89]. *NTRK* inhibitors are well-tolerated, and most adverse events are grade 1–2, including fatigue, nausea, neurologic events (cerebellar ataxia, cognitive disorder, and dizziness), weight gain, and anemia [62,87,88,89]. Both entrectinib and larotrectinib have tumor agnostic FDA approval for *NTRK* rearranged tumors.

### 4.8. NRG1 Fusions

Neuregulin 1 (*NRG1*) is a soluble ligand that ordinarily is released from cells and acts as a paracrine messenger [90,91]. *NRG1* binds to HER3, promoting HER2/HER3 dimerization and PI3K/AKT/mTOR signaling [90,91]. *NRG1* fusions are oncogenic drivers in PDAC and account for 1.3% of *KRAS* WT cases [63,92]. Mechanistically, *NRG1* fusion proteins tether the *NRG1* to the plasma membrane, resulting in hyperactivation of HER3 [90,91]. Zenocutuzumab, an investigational bispecific antibody targeting HER2 and HER3, has shown promising efficacy in patients with *KRAS* WT PDAC with *NRG1* fusions [93]. In a phase II clinical trial investigating zenocutuzumab monotherapy, there was a 42% ORR in the 19 patients with *KRAS* WT PDAC patients harboring a *NRG1* rearrangement [93]. In all patients, median time to response was 1.8 months and duration of response was 9.1 months. The toxicity profile of zenocutuzumab was generally favorable, although there was one grade 5 hypersensitivity reaction. The most common grade ≥ 3 adverse events were fatigue (4%), dyspnea (4%), anemia (3%), and liver test abnormality (3%). Afatinib, a pan-ERBB inhibitor, is also being explored in *NRG1* fusion-associated PDAC [63,92]. There have been published reports of two patients with PDAC with *NRG1* rearrangements responding to afatinib, and clinical trials of this agent are ongoing [63,92].

### 4.9. RET Fusions

The *RET* proto-oncogene encodes for a transmembrane receptor tyrosine kinase involved in embryonic development of the nervous system and kidneys [94]. *RET* fusions lead to aberrant activation of *RET* receptor tyrosine kinase and constitute 1.3% of *KRAS* WT PDAC [29]. In the LIBRETTO-001 phase I/II basket trial, 12 patients with PDAC harboring a *RET* rearrangement were enrolled and treated with the *RET*-specific inhibitor selpercatinib [95]. The patients on the trial had a 55% ORR, with a range of response duration from 2.5 months to 38.3 months. Approximately 38% of patients had at least one grade 3 adverse event (AE), which included mostly liver test abnormalities. In the ARROW phase I/II clinical trial investigating the *RET* inhibitor pralsetinib in solid-tumor patients with RET rearrangements, nine patients with PDAC were enrolled [94]. All four patients who were evaluable had an objective response (3 PRs and 1 CR with a treatment duration of 33.1 months) [94]. In the evaluable group of all solid tumor patients, the ORR was 57%, with a median duration of response of 11.7 months. On the trial, 69% of patients experienced grade ≥3 treatment-related adverse events. The most common grade ≥3 toxicities were neutropenia, anemia, and liver test abnormalities requiring dose interruptions (59%) or dose reductions (45%). Selpercatinib has been granted a tumor agnostic FDA-approval for treatment of RET-fusion positive tumors.

### 4.10. ROS1 Alterations

The *ROS1* gene encodes for a receptor tyrosine kinase that is closely related to *ALK*. It is a transmembrane protein with an intracellular tyrosine kinase domain and large extracellular domain [96]. *ROS1* fusions are rare subsets of *KRAS* WT PDAC accounting for less than 1% of all patients and published reports on the efficacy of targeting *ROS1* in PDAC are extremely rare [27,29]. There has been one case report of a patient with *KRAS* WT PDAC who had stable disease on entrectinib for 7 months before progression of disease [62]. Due to its rare incidence, no other larger cohorts have been reported. No pancreatic cancer specific FDA approved therapy is available for *ROS1* altered PDAC.

### 4.11. MMR Deficient PDAC

Mismatch repair deficient (MMRd) PDAC accounts for up to 2.5% of all PDAC and 4.7% of KRAS WT PDAC [29,97]. Often, MMRd PDAC is frequently TP53 WT and often has a medullary or mucinous/colloid histology [98]. In an early phase clinical trial studying pembrolizumab, eight MMRd PDAC patients were enrolled; five of these achieved a partial response to therapy (62% ORR) [99]. However, in the KEYNOTE-158 study, there was a more modest 18.2% (*n* = 4/22 patients) ORR in MMRd PDAC who had previously had progression on first-line therapies [100]. There were no unexpected immune-related adverse events in these trials, with 14.6% experiencing grade ≥ 3 adverse events, including liver test abnormalities (GGT and ALT), pneumonitis, Guillain Barre Syndrome, neutropenia, and autoimmune enterocolitis. Pembrolizumab has a tumor-agnostic FDA-approval for MSI-H/MMRd tumors, which allows patients with MMRd PDAC access to this promising therapy. While pembrolizumab is a recommended therapy for MMRd PDAC, there is currently no role for immunotherapy in microsatellite stable/mismatch repair proficient PDAC, as evidenced by multiple negative clinical trials investigating its use [60,101]. However, future investigations are trying to identify methods of overcoming immunotherapy resistance in mismatch repair-proficient PDAC.

## 5. Conclusions

PDAC is a highly lethal malignancy, and patients are in desperate need of molecularly targeted therapies that can improve the current treatment paradigms (Figure 2). Cytotoxic chemotherapy remains the standard of care for most patients with PDAC. While combination chemotherapy regimens have improved outcomes in the first-line setting, the efficacy of second-line chemotherapy is modest. The use of next-generation sequencing (NGS) can identify patients who might benefit from targeted therapy. FDA-approved options are available for PDAC tumors with *BRCA*1/2 mutations (germline), MMRd, *NTRK* fusions, *BRAF* V600E mutations, and *RET* fusions. Investigational therapies, as well as off-label use of therapies that have FDA approval for other malignancies, are available to *KRAS* WT PDAC patients harboring other *BRAF* alterations and *ALK*, *FGFR2*, *ROS1*, and *NRG1* translocations. However, there is a large portion of PDAC patients who do not have a targetable alteration, and innovative targeted and immunotherapies that can improve outcomes in these patients are greatly needed.

## Figures and Tables

**Figure 1 cancers-14-06223-f001:**
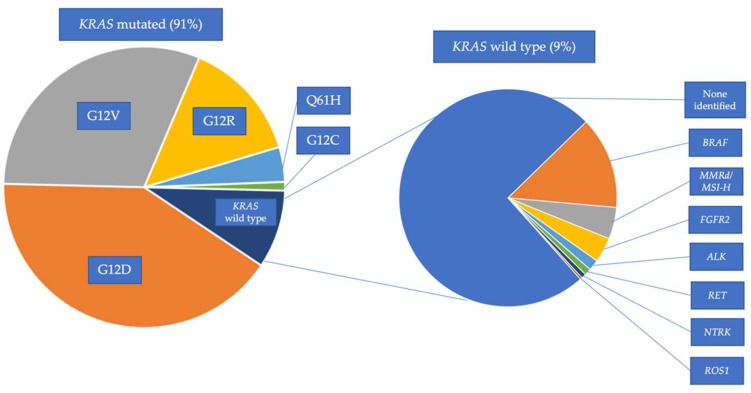
Oncogenic driver alterations in *KRAS*-mutated and *KRAS* wild-type PDAC. The proportion of *KRAS* mutations in PDAC include *KRAS* G12D (41%), G12V (31%), G12R (14%), Q61H (4%), G12C (1%), and *KRAS* wild-type (WT) (9%) [29,30]. Actionable alterations in *KRAS* WT patients in PDAC and pancreatic acinar cell carcinoma include: *BRAF* alterations (13.7%), MMRd/MSI-H (4.7%), *FGFR2* fusions (3.5%), *ALK* fusions (1.6%), *RET* fusions (1%), *NTRK* fusions (0.7%), and *ROS1* fusions (0.3%). Frequencies of genomic alterations in *KRAS* WT were derived from Philip et al. and Singhi et al. [29,30].

**Figure 2 cancers-14-06223-f002:**
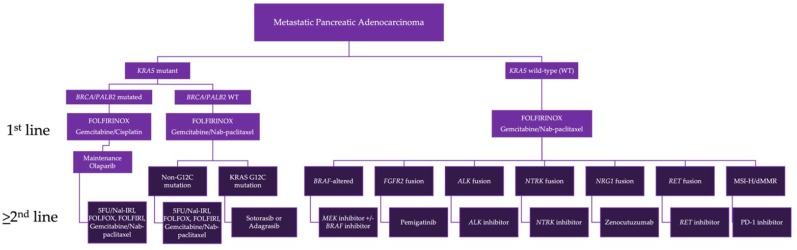
Biomarker-driven treatment paradigm for metastatic pancreatic adenocarcinoma. Treatment algorithm describes first-line, maintenance, and second-line treatment options for PDAC patients, based upon the tumor’s molecular alterations. Multiple treatment options are listed in instances where more than one therapy is available. Abbreviations used include: FOLFIRINOX: folinic acid, 5-fluorouracil, irinotecan, and oxaliplatin; FOLFOX: folinic acid, 5-fluorouracil, and oxaliplatin; Nal-IRI: nanoliposomal irinotecan; FOLFIRI: folinic acid, 5-fluorouracil, and irinotecan; MSI-H: microsatellite instability-high; dMMR: mismatch repair deficient; WT: wild type.

**Table 1 cancers-14-06223-t001:** Recommended FDA-approved targeted therapy options for PDAC.

Recommended FDA-Approved Targeted Therapies in PDAC	
**Molecular Target**	**Drug Name**
g*BRCA1/2* mutations	Olaparib
**Tumor agnostic FDA-approved targeted therapies**	
**Molecular Target**	**Drug Name**
MSI-H/MMRd	Pembrolizumab
BRAF V600E	Dabrafenib/Trametinib
NTRK fusions	Larotrectinib, Entrectinib
RET fusions	Selpercatinib
**Therapies that are FDA-approved in other tumors**	
**Molecular Target**	**Drug Name**
*KRAS* G12C	Sotorasib, Adagrasib
*FGFR2* fusions	Pemigatinib
*ALK* fusions	Crizotinib, Alectinib
*ROS1* fusions	Entrectinib, Crizotinib
s*BRCA1/2*, g*PALB2*/s*PALB2* ^1^	Niraparib, Olaparib, Rucaparib

^1^ s: somatic; g: germline.

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
