# Peer review of "Emerging Role of Targeted Therapy in Metastatic Pancreatic Adenocarcinoma"

_cancers, 2022, doi:10.3390/cancers14246223_

Round 1

Reviewer 1 Report

The manuscript "Emerging Role of Targeted Therapy in Metastatic Pancreatic Adenocarcinoma” by Huffman et al. offers to the reader a usefully organized, well written and comprehensive overview of the current available options for metastatic PDAC treatment. The manuscript is relevant to a large audience, including both scientists and clinicians, because of the primary interest in cancer precision medicine.

In my opinion, the addition of more information about preclinical promising therapeutic approaches based on gene or cell therapy (briefly mentioned only in the section KRAS mutated PDAC) as well as combinations of drugs targeting not only the oncogene driver but also the microenvironment would further improve the paper.

Author Response

We greatly appreciate the reviewer's thoughtful feedback. We agree that the suggested topics are very interesting and important. However, we aimed to provide a clinically oriented review of reported clinical data of the use of targeted therapies in pancreatic cancer, and we believe that the suggested additions of preclinical evidence of combinations, microenvironment therapies and gene/cell therapies are very important, but beyond the scope of this article.

Reviewer 2 Report

The current manuscript reviews targeted therapies in metastatic pancreatic cancer. This is an interesting and timely topic. The review is up-to date and well written. The relevant literature is cited and discussed. I have no further comments or suggestions. Congratulations on an interesting piece of work.

Author Response

Thank you very much for your comments.